# Modeling and Simulation of a Low-Cost Fast Charging Station Based on a Micro Gas Turbine and a Supercapacitor

Bogdan Gilev [1], Miroslav Andreev [2], Nikolay Hinov [3,*] and George Angelov [2]

1 Department of Applied Mathematics, Technical University of Sofia, 1756 Sofia, Bulgaria
2 Department of Microelectronics, Technical University of Sofia, 1756 Sofia, Bulgaria
3 Department of Power Electronics, Technical University of Sofia, 1756 Sofia, Bulgaria
* Correspondence: hinov@tu-sofia.bg; Tel.: +359-29652569

**Abstract:** In recent years, micro turbine technology has become a continuously reliable and viable distributed generation system. The application of distributed energy power generation sources, such as micro gas turbines (MGT), to charge electric vehicles offers numerous technical, economical benefits, and opportunities. MGT are considered as they are smaller than conventional heavy-duty gas turbines. They also are capable of accepting and operating with different fossil fuels in the range of low–high pressure levels as well as co-generation opportunities. The MGT could provide the fast and reliable output power guaranteed and needed for grid stability. This paper provides a mathematical representation, modelling, and simulation of a low-cost fast charging station based on a micro gas turbine and a super capacitor forming altogether a power generation system suitable for use especially as energy source in fast charging stations and dynamic power systems. All the micro gas turbine's parameters are estimated according to available performance and operational data. The proposed system generates up to 30 kW output power assuming that it operates with natural gas. The developed model of the system is simulated in the environment of MATLAB/Simulink. Each part of the micro turbine generation system is represented by a mathematical model. On the basis of the developed model of the system, the minimum value of the supercapacitor was determined, which ensures the charging schedule of a selected electric vehicle.

**Keywords:** combined heat and power (CHP) technology; combustion chamber; compressor hybrid electric vehicle; control system; distributed generation system; micro gas turbine (MGT); permanent synchronous magnet machine (PMSM); power converter

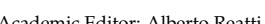



## 1. Introduction and Motivation

Micro gas turbines (MGT) have become more popularity nowadays in distributed and co-generation systems. They have many strengths and benefits, such as compact small size, high speed turbo-generators, low noise, low emissions characteristics, and high efficiency, by using a recuperator with much faster response than the conventional turbines [1–3]. Currently, there is growing interest in the use of MGTs for on-peak power supply for supporting the power grid. They represent small combustion turbines which have output ranges from 20–30 kW to 400–500 kW and are able to generate energy in different forms, such as electricity and heat [2,3]. They are also capable of accepting and operating with different fossil fuels in the range of low–high pressure levels as well as co-generation opportunities. The MGT could provide fast and reliable output power guaranteed and needed for grid stability.

The spread and application of MGTs is due to the continuous progress in the field of power electronics, which allow their sustainable operation and interaction with the electrical network. Using power electronic converters for power management, the synchronous machine (generator) does not need to be synchronized with the power grid. On the other hand, it allows the generator and the turbine to be integrated together on a common shaft [2–5].

Following the decentralized power generation scenario incorporating micro gas turbines and other power energy sources, such as wind turbines, PV systems, fuel cells, and energy storage (battery and supercapacitors) will definitely provide a high efficient, economically viable, and environmentally friendly on-site power energy system which can be connected close to the consumer load, providing the end customer with heat and electricity without any conversion losses. Electric vehicle fast chargers can be implemented and integrated at the remotely located gas stations using micro gas turbines for energy generation [5,6]. A very important advantage of micro turbines in relation to other heat elements used for decentralized generation systems is their capability to use a variety of gaseous, i.e., natural gas, biogas (landfill, digester), sour gas, propane gas, and liquid, kerosene, aviation fuels.

The MGT will play an essential role in the future energy power scenario in order to guarantee the security of power supply and achieve a sustainable, cost-efficient energy generation system. MGTs have some advantages, e.g., fast response to the changes occurring in the load, low harmful emissions, low fuel consumption rate, strong fuel adaptability, low maintenance cost, and higher stability than internal combustion engine driven generators, and are more preferred rather than diesel groups for autonomous or backup power supply. Some of components taking part in MGT system cannot exist in a particular MGT and it depends on the requirements of the system and application. The small size of the MGT power generation system is a big advantage because of these technologies use high-speed turbines in range from 50,000 to 120,000 RPM with air bearings [7,8].

One of the main challenges to the introduction of the various types of electric vehicles (hybrid electric vehicles, HEVs; plug in hybrid electric vehicles, PHEVs; and pure electric vehicles, PEVs) is the development of the charging infrastructure and the related sustainable production of electrical energy. The use of PV and wind turbines is related with many difficulties, because of the stochastic nature of the energy they produce [9]. On the other hand, the construction of charging stations powered entirely by the power grid leads to a significant increase in energy consumption and to problems related to sustainability. In this aspect, the use of hybrid stations is proposed, which are powered both by the grid and by decentralized energy production, based most often on photovoltaic generators (PV). A new type of charging station (CS) integrated with a renewable energy source is proposed in [10]. By using multi-criteria optimization, the optimal power of the CS was found, taking into account many factors: economic, technical, environmental (reduction of emissions), green energy production possibilities, and the size of the energy storage element. In [11], the design and prototyping of a hybrid charging station for electric vehicles is discussed. The CS is jointly powered by solar energy and the electricity grid. The energy flow management system optimizes the use of energy from the grid. It is designed as follows: it takes energy from solar panels and directly charges the EV when solar energy is available. When no solar energy is available, the system is powered by the grid. On the other hand, to increase the efficiency of using solar energy, the system supplies solar energy to the grid when solar energy is available, but there is no EV connected to the CS. In this mode, the CS will operate as a grid-connected solar power plant. In [12], a bi-directional DC CS using a blocking capacitor-based DAB converter and an active forward converter is presented. The system is designed to meet the requirements of power factor, lower harmonics, and deep power regulation standards. The developed system provides bi-directional power flow for a 40-kW power supply. In [13], a simple configuration of a hybrid charging station is proposed for EV fast charging, aiming to reduce the impact on the grid. A detailed PV model with a grid-integrated EV charging station using a multi-port power electronic converter is considered. A power flow control is synthesized to ensure both the stabilization of the power grid and the charging profile of the EV. An additional advantage of the presented structure is the reduction of peak power and network load balancing. The applicability and efficiency of the proposed multiport converter is evaluated using MATLAB/Simulink. In [14], the development and implementation of Ultra-Fast DC charging station technology is reviewed. This technology has great development potential and is very attractive to EV drivers thanks

to its fast charging speed of 10 min. The main aspects and limitations to the implementation of this cutting-edge technology are discussed in detail: the required available power of the fast charging station, advanced energy conversion technologies based on modified and new electronic converter power circuits, upgradeability and modular development, impact on the power grid, impact on the level of EV penetration, development of a suitable business model and price tariffs. The research is very useful in view not only for EV users, but also for grid system operators because it provides effective solutions to reduce costs, increase profit and integrate a large number of EVs into the grids. As a summary of the leading research on the subject, it should be concluded that the trends for the development of CS are related to a reduction in charging time, which leads to an increase in the installed capacities and the peak load of the networks. In this aspect, one of the possible solutions for providing electrical energy with good economic, ecological, and technical parameters are MGT.

The above advantages of MGT give a good opportunity to use them to create their base of charging stations for electric vehicles, including those for fast charging. [6,15].

All electric vehicle propulsion technologies are based on the use of energy sources, such as batteries, supercapacitors, fuel cells, or hybrid energy storage systems involving several components. In the implementation of charging stations, in order to be able to improve the dynamics and not to disturb the stability of the electrical grid (especially with micro- or nano-grids), it is necessary to use an energy storage element to ensure the necessary load schedule. In this aspect, to ensure the operability of the charging station, a supercapacitor and/or storage battery (with the corresponding energy flow management system) is added to the MGT for an energy buffer while the micro gas turbine power is adapted to the change in load.

The main purpose of the manuscript is to study a system for the fast charging of electric vehicles which is autonomous from the power supply network, but at the same time consists of a minimum number of power electronic devices and an energy storage element (supercapacitor) with the smallest possible capacity. In this way, the system will be low-cost and therefore competitive with other options for charging stations. In this regard, the aim of the authors is to prove with the method of mathematical modelling the feasibility and possibilities for using this type of system to expand the possibilities for using electric vehicles. The developed and simulated model of the system is built in the MATLAB/Simulink environment.

## 2. System Overview and Modelling of a Micro Gas Turbine Generation System with PMSM

### 2.1. System Overview of the Power Generation Based on a Micro Gas Turbine with a High-Speed Generator

In this study, a model of a single shaft micro gas turbine based on the Capstone micro turbine [16,17] is developed and simulated. The main components that this power generation system consists of are a turbine generating the mechanical power, high-speed permanent magnet synchronous machine generator (PMSM) delivering the electric output power, a compressor, a heat recuperator (HR) allowing the system to recover thermal energy and increase the efficiency of the system accordingly, a combustion chamber (CC) allowing and accepting the burning of different types of fuel, and a three-phase power electronic AC/DC converter (non-controllable) allowing the integration of micro gas turbine with DC grids and energy storage systems, e.g., supercapacitors etc. In the specific case, the DC bus is composed of a one-way DC/DC converter, through which the regulation of the power supplied by the microturbine to the load is carried out, and a bi-directional DC/DC converter, through which the energy exchange between the supercapacitor and the DC bus takes place. If there is a need to supply AC loads, a three-phase inverter is added to this structure.

Figure 1 shows the investigated structural diagram of the charging station based on MGT and supercapacitor. The description of the principle of operation of the presented structure is as follows: The micro turbine drives the permanent magnet synchronous

machine (PMSM) at a high speed level, typically 96,000 rpm, and generates a high frequency variation, i.e., 1500–4000 Hz. The high-frequency AC voltage is then rectified. The control of the output power generated by the microturbine and given to the battery of the electric vehicle is carried out by the one-way DC/DC converter. The supercapacitor provides the necessary energy until the MGT establishes its operating mode. Once the required power is reduced depending on the battery charge cycle of the electric vehicle, then the supercapacitor is charged to be ready for the next start of the system. Since supercapacitors are usually composed of stacks with a much lower voltage than that of the DC bus, the bidirectional DC/DC converter is step-up (Boost) in the direction of energy transfer from the supercapacitor to the DC bus and step-down (Buck) in the opposite direction. Pulse width modulation (PWM) control is usually used to control the DC/DC converter and the turbine can also rely on mechanical stabilization of its revolutions by means of the fuel supply system. In the case under consideration, an MGT with a recovered compressor and a turbine connected to the same shaft and powered by natural gas was used. MGT works based on a thermodynamic cycle known as the Brayton cycle. The air stream is introduced into the compressor and compressed through multiple stator and rotor stages. The compressed air is then mixed with fuel in the combustion chamber where this process takes place. As a result, the hot gas is expanded through several turbine stages to drive the generator and compressor [18–20]. Thus, if operation in cogeneration or trigeneration mode is used, a very high efficiency of the entire system is achieved, and the released heat can be used either for heating or cooling, depending on the climatic season and the location of the charging station. This is reflected in the block diagram of the microturbine, which is presented in Figure 1.

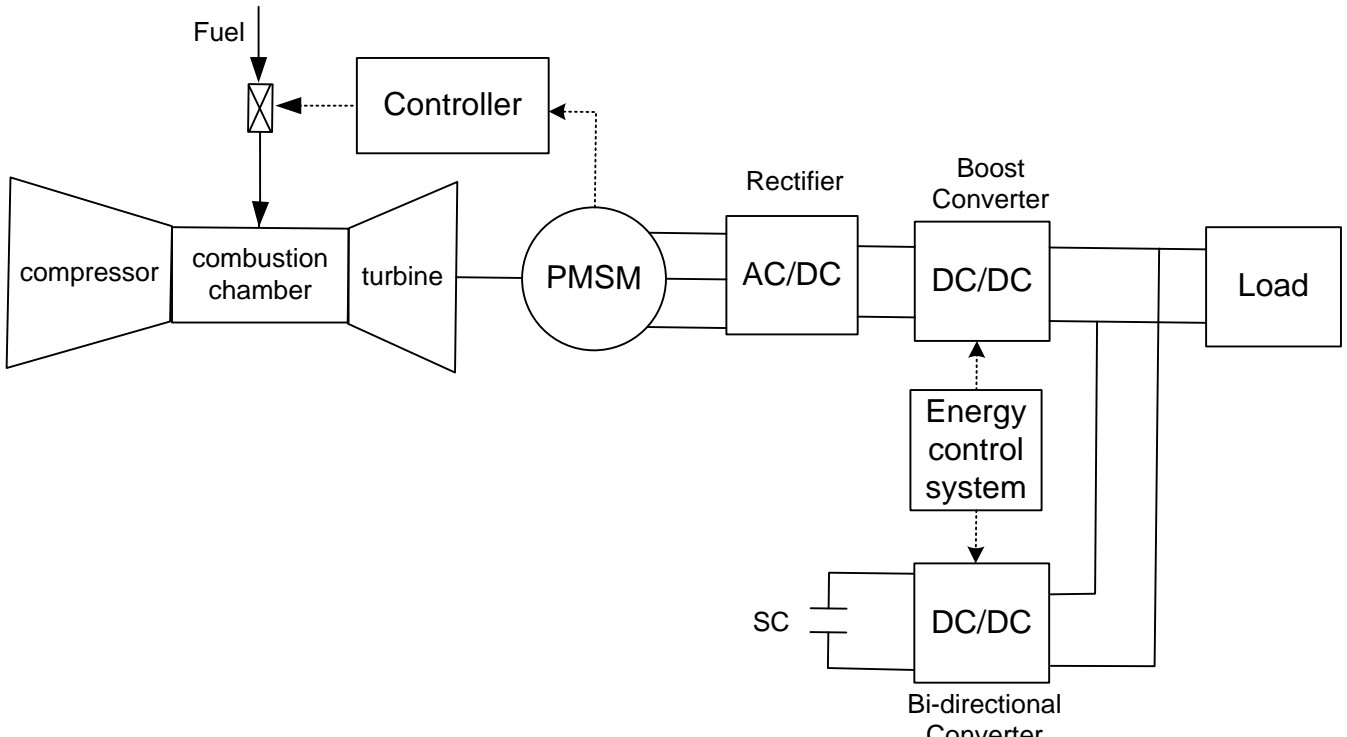

**Figure 1.** A block diagram of a charging station based on MGT and a supercapacitor.

In this paper, a single-shaft micro-turbine single-shaft is modeled based on Rowen's model [21]. The model of the micro gas turbine is implemented using Simulink/MATLAB and a block diagram control system of the micro gas turbine, as shown in Figure 2.

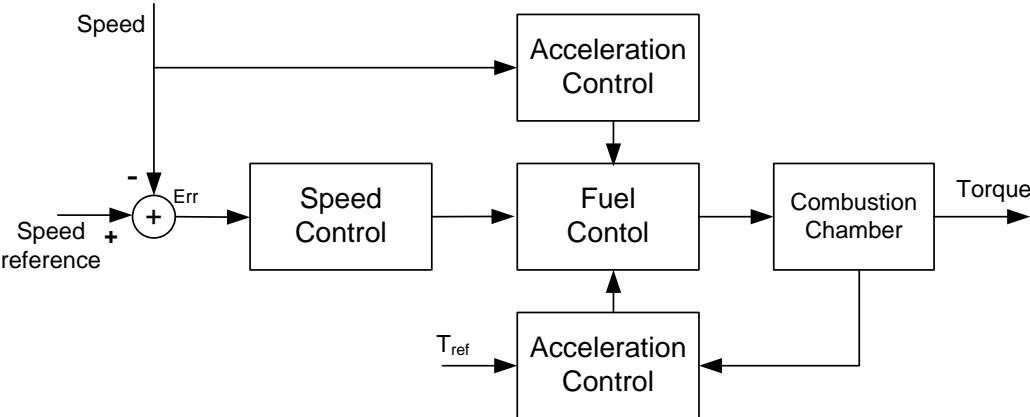

**Figure 2.** Block diagram of control system of the micro gas turbine.

The Simulink model of the MGT used for modelling the charging station is shown in Figure 3. Due to their wide use in the energy industry, numerous MGT models have been developed and implemented in different software environments, such as [20,22–25]. Usually, these models take into account only some specific processes in the turbine or are more complex, thus not allowing the achievement of large simulation times (hundreds and thousands of seconds) using standard computer architectures, such as those available in universities and research centers. From a literature review, it was found that Rowen's model gives good accuracy while using relatively few computational resources. The simulation parameters of the micro gas turbine are given in Appendix A.

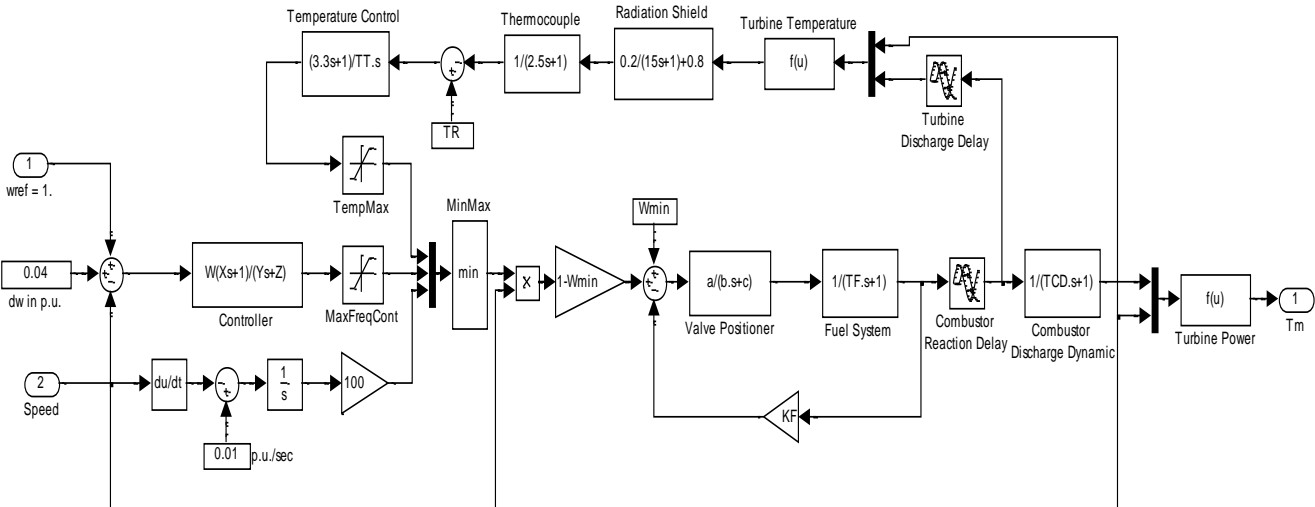

**Figure 3.** Simulink model presenting realization of simplified single-shaft gas turbine for isolated operation.

Generally, the gas turbine model based on Rowen uses per unit values, except for the temperatures. For this, when this model is added to the model of the whole system, two amplifier blocks are added before and after it, which convert from SI to PU units at the input and from PU to SI units at the output. The actual equivalents of the unit values for the various machines are different. Therefore, their direct use in reference equations would decrease the universality of the model.

The model is based on two control systems of fuel process, two valves in series. The control systems perform the following functions: speed control, temperature control, and acceleration control, and based on these three control systems the lower and upper fuel limits are determined, as shown on Figure 2.

The first system has three main functions control: a speed control under partial load conditions, temperature control acting as an upper limit, and acceleration control to prevent

exceeding of the speed acting as a lower limit. The speed control function reaches out to reduce the difference between the reference speed and actual speed of the system or rotor. The temperature control function is fully used to reduce and limit the power of the turbine at a pre-defined set point of the firing temperature regardless of changes in ambient temperature or fuel characteristics. The acceleration control is mainly used during the start-up process of the turbine for limitation of the acceleration rotor speed before reaching the regulator speed. Thus, it can improve the thermal stresses which can be generated during the start-up process. This control function executes a secondary role during normal operation. The output of the selector with the lowest values (called VCE) is the lowest from the three inputs and results in the least amount of fuel to the compressor-turbine [26].

As mentioned, the fuel gas control system consists of two valves in series, the first of which drives the pressure between the two valves depending on the speed. The second valve is designed so that the velocities in the controlled area have a pressure ratio of up to 1.25. If the valve position is maintained in proportion to the VCE signal, the rate of fuel gas flow is proportional to the VCE output and the turbine speed.

There are only two important time constants in this system. The first is connected to the positioning system of the gas control valve and the second is the volume time constant associated with the pipelines and the fuel gas distribution manifold. Therefore, the isolated micro gas turbine is essentially a linear, non-dynamic device except for the time constant of the rotor.

### 2.2. Modelling of the PMSM (Permanent Magnet Synchronous Machine, High-Speed Generator)

The used PMSM in the proposed model is a commonly two pole PMSM generator with sinusoidal magnetic field [26–28]. In the PMSM model, some of the main assumptions have been considered, such as electrical asymmetry, magnetic asymmetry, and the magnetic flux being sinusoidal distributed along the air gap and having no saturation. Mathematical equations of PMSM are presented to the synchronous d-q rotating frame by aligning the d-axis with the direction of the rotor flux. The interfacing voltages $u_{ab} = u_a - u_b$ and $u_{bc} = u_b - u_c$ are reduced to $qd$-axis by means of the Park transformation.

$$\begin{pmatrix} u_q \\ u_d \end{pmatrix} = \frac{2}{3} \begin{pmatrix} \cos(\theta_e) & -\sin(\theta_e) \\ \sin(\theta_e) & \cos(\theta_e) \end{pmatrix} \begin{pmatrix} 1 & 1/2 \\ 0 & -\sqrt{3}/2 \end{pmatrix} \begin{pmatrix} u_{ab} \\ u_{bc} \end{pmatrix} \tag{1}$$

where $\theta_e = p\theta$ is an angle of rotation of the electromagnetic field, $\theta$ is a mechanical angle of rotation of the rotor, and $p$ are pairs of poles ($p = 4$).

Calculate the current values of $i_q$ and $i_d$

$$\begin{aligned} \frac{di_q}{dt} &= \frac{1}{L_q} \left( u_q - R\,i_q - L_d \omega_e i_d - \Phi \omega_e \right) \\ \frac{di_d}{dt} &= \frac{1}{L_d} \left( u_d - R\,i_d + L_q \omega_e i_q \right) \end{aligned} \tag{2}$$

where $\omega_e = p\omega$ is the angular speed of the electromagnetic field, $\omega$ is the mechanical angular speed of the rotor, $p$ are pairs of poles ($p = 4$), $\Phi$ is magnetic flux $\Phi = 0.175\ Wb$, and $R$ is resistance $R = 2.875\Omega$.

$$L_q, L_d \text{ are inductances } (L_q = L_d = 8.5\ mH) \tag{3}$$

The values of these parameters for the modeled PMSM were determined using the methodology presented in [29].

In order to compute the stator currents, it is necessary to use inverse Park transformations.

$$\begin{pmatrix} i_a \\ i_b \end{pmatrix} = \begin{pmatrix} 1 & 0) \\ -1/2 & -\sqrt{3}/2) \end{pmatrix} \begin{pmatrix} \cos(\theta_e) & \sin(\theta_e) \\ -\sin(\theta_e) & \cos(\theta_e) \end{pmatrix} \begin{pmatrix} i_q \\ i_d \end{pmatrix} \text{ and } i_c = -i_a - i_b \tag{4}$$

With the calculated electric magnitudes, the electric torque of the motor can be found.

$$T_e = 1.5 \, p\big(\Phi \, i_q + (L_d - L_q)i_d i_q\big) \tag{5}$$

The torque obtained in the above equation is involved as an external influence in the next equation.

The mechanical part of the electric motor and the other rotating mechanical parts are modeled together. For this purpose, the torque of the other rotating parts is brought to the torque of the electric motor shaft ($J = J_{em} + J_{gt}/k_{gear}^2$) and a single-mass model is obtained. This model is described by the following differential equations system.

$$\begin{aligned} \frac{d\omega}{dt} &= \frac{1}{J}\big(-T_m + T_e - F\omega_r\big) \\ \frac{d\theta}{dt} &= \omega \end{aligned} \tag{6}$$

where $\omega$ is an angular speed of the rotor, $\theta$ is a mechanical angle of rotation of the rotor, $J$ is a is the equivalent inertia to the rotor, $F$ is a friction coefficient, $T_e$ is electric torque, and $T_m$ is a mechanical torque.

## 3. Model of the Entire System

The modeled overall charging station is shown in Figure 4. The rated voltage of micro gas turbine (line to line) is 360 V and the maximum output power reached is about 30 kW. Micro gas turbine connects with load via AC-DC rectifier and Boost DC/DC converter. Different load profiles (charging cycles) of electric vehicle Nissan Leaf are presented in Figure 5. The load mode 2 is realized in the "load" block in model. The gas turbine speed reference value is set to 1 p.u. in whole simulation process.

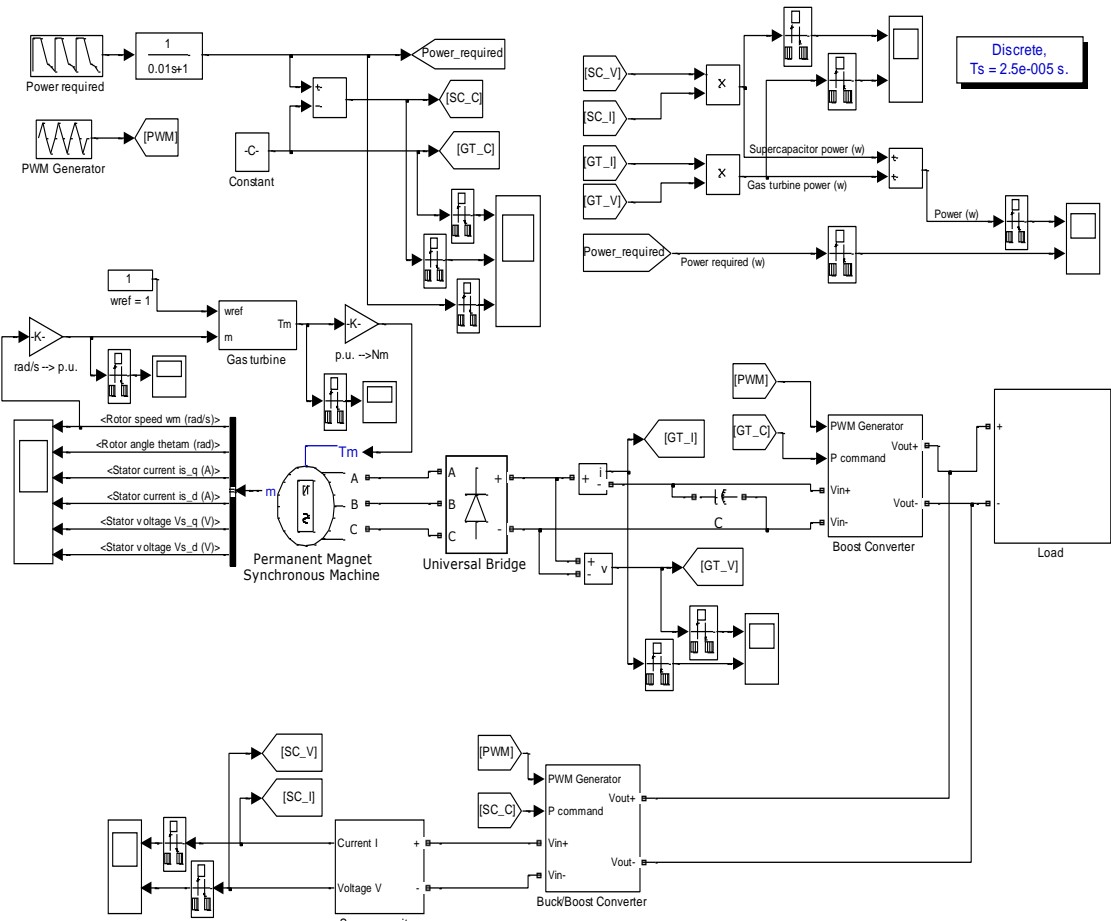

**Figure 4.** The model of the complete electric vehicle charging system based on MGT and SC.

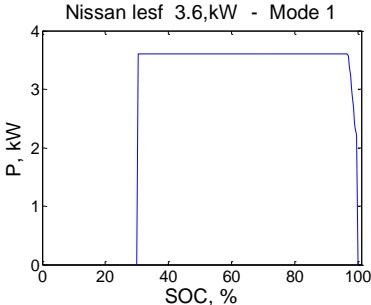
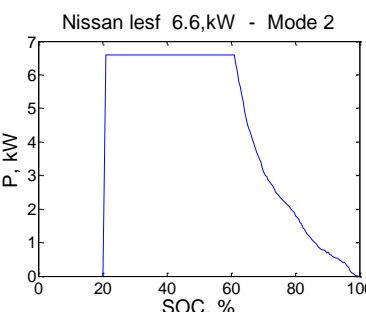
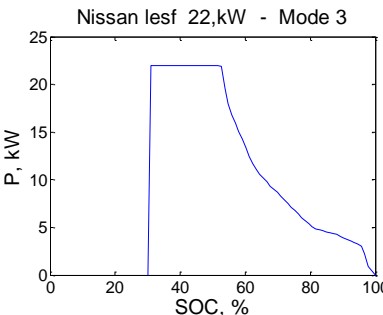

**Figure 5.** Different charging cycles of Nissan Leaf.

The strategy of the control system is to keep the operation of the MGT close to the rated power and with the help of super capacitor (SC) to compensate for the shortage/excess of energy during the charging process. For this purpose, the SC is connected in parallel with the MGT-PMSM module by bidirectional DC-DC converter. The direction of energy transfer is controlled by two DC/DC converters (boost converter, bi-directional converter). Because the SC can be both a consumer and a source of energy, it is controlled by a bi-directional converter. The MGT-PMSM module is only a source of energy, and it is controlled by a boost converter. The built-in transistor models from the MATLAB/Simulink environment were used in the modeling of the DC/DC converters. In [30,31], the effects of parasitic elements on the dynamics and characteristics of DC/DC converters are presented, and in [32], a methodology for identifying and modeling parasitic components in power electronic devices is given. The present research is not focused on these aspects, especially since in the cited manuscripts it was found that the introduction of the parasitic elements reduces the jumps of the output voltage and current of the converters during the transient processes, and the reduction of these established values is compensated by the controller.

The charging cycles of the electric vehicle were taken from the website of the manufacturer [33], and by using a model-based approach for designing the charging station, the parameters of the power electronic converters and their management are adapted to the specific needs and requirements.

The MATLAB/Simulink implementation of the bi-directional DC/DC converter is shown in Figure 6. A classic power circuit with two transistors and their corresponding reverse diodes was used in the modeling. The control system is required to reverse the direction of energy from the DC bus to the supercapacitor and vice versa depending on the energy needs of the load. This is implemented as follows:

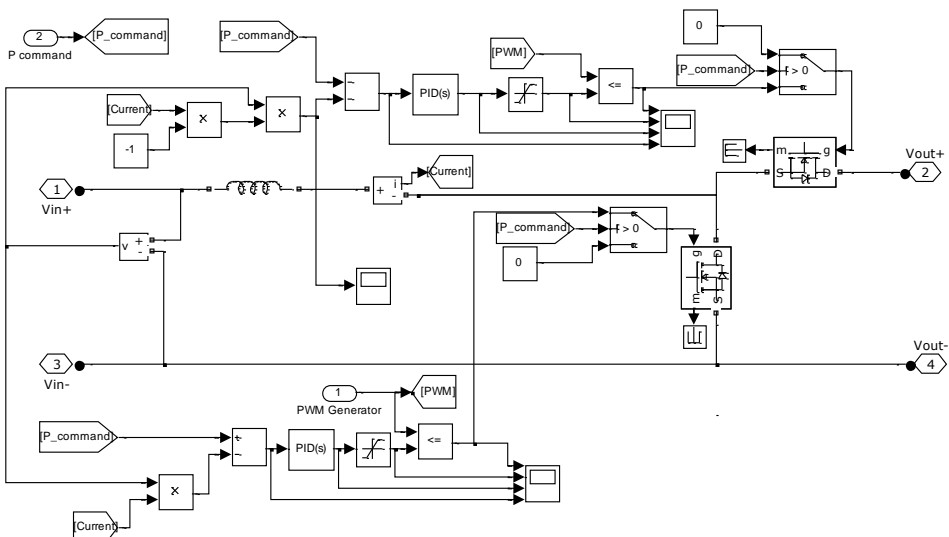

**Figure 6.** Model of bi-directional DC/DC converter.

The reference power (load schedule) that the system must provide, by using mathematical functions, is divided into two parts: first, a constant power close to the nominal one that PMSMachine generates, and second, power obtained by subtracting this constant power from the reference. The first component of the task (providing constant power in the load) is fed as a reference to the boost converter (Figure 6). The second component of the task (corresponding to the variable component of the power) is fed to the control input of the bi-directional converter (Figure 5). When the variable component of the power is positive, a control signal 1 is applied to the transistor connected in series to the Vout+ bus, and a control signal 0 is applied to the parallel transistor (and it works as a diode) and the bi-directional converter operates in the step-down mode, providing charging of the SC. When this variable power component is negative, a control signal 1 is applied to the parallel connected transistor and a control signal 0 is applied to the transistor connected in series to the Vout+ bus (it works as a diode) and the bi-directional converter operates in step-up mode as the discharge of the SC, provides the missing power of the load. Transistor control pulses are generated with PID controllers and a PWM system (Figures 5 and 6).

## 4. Simulation Results and Finding the Optimal Value of Supercapacitor SC

In the model experiments, the load current was selected according to the Mode 2 charging cycle, 6.6 kW at an initial state of charge (SOC) of 30%. Since the supercapacitor (in practice, it is a stack of individual elements) is an expensive element, the purpose of these experiments is to determine its minimum value that ensures the system's operability. This value is found by running a series of simulations with the model. Initial simulations are performed with values of 300 F and 500 F. Figures 7 and 8 show the results of the simulated overall system. Obviously, at C = 500 F (Figure 8), the realized and reference power are very close, while at the smaller value of 300 F (Figure 7) the charging station is not able to perform the power assignment.

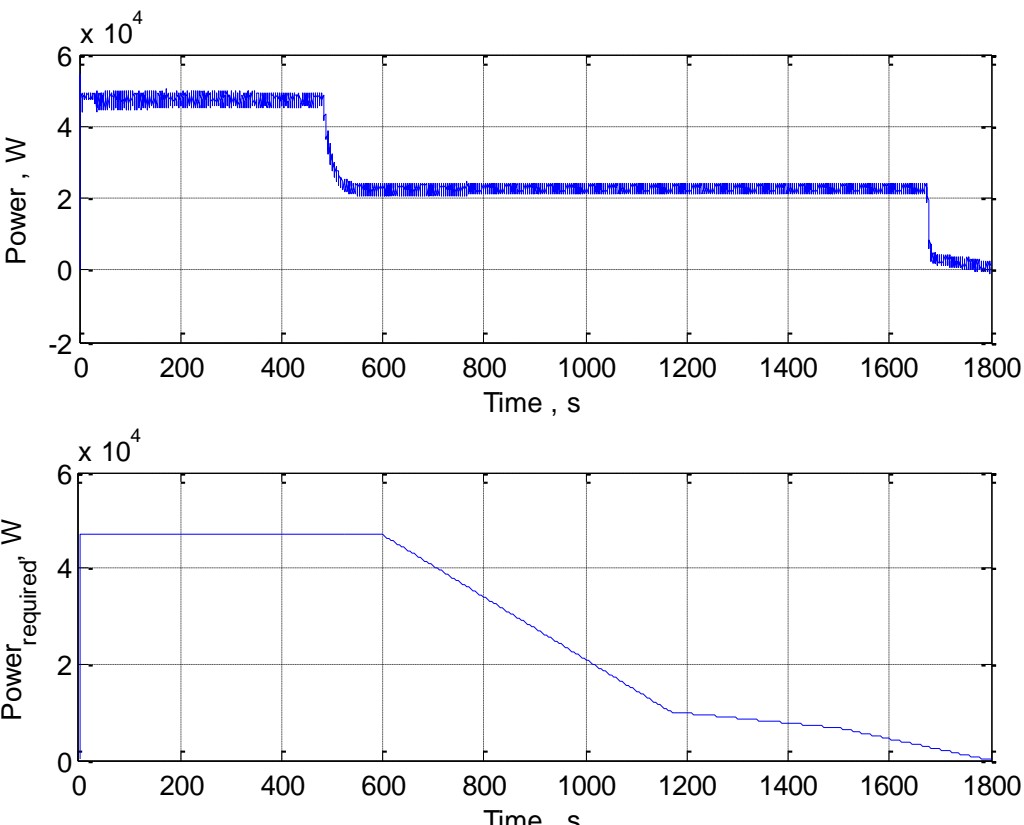

**Figure 7.** Realized and reference power at C = 300 F.

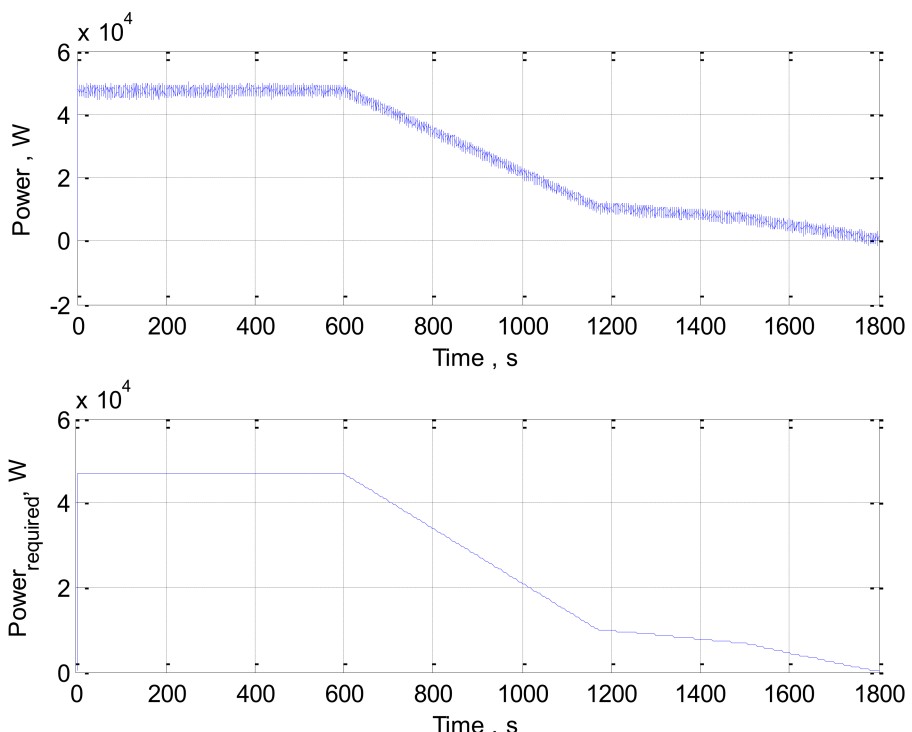

**Figure 8.** Realized and reference power at C = 500 F.

Then, by the method of bisection, the minimum possible value of SC for which realized and reference power remain very close is found. This value is C = 481 F (Figure 9). A suitable supercapacitor battery that meets the requirements would be [34]. With this optimal value, the results of the following Figures 9–12 are simulated and presented.

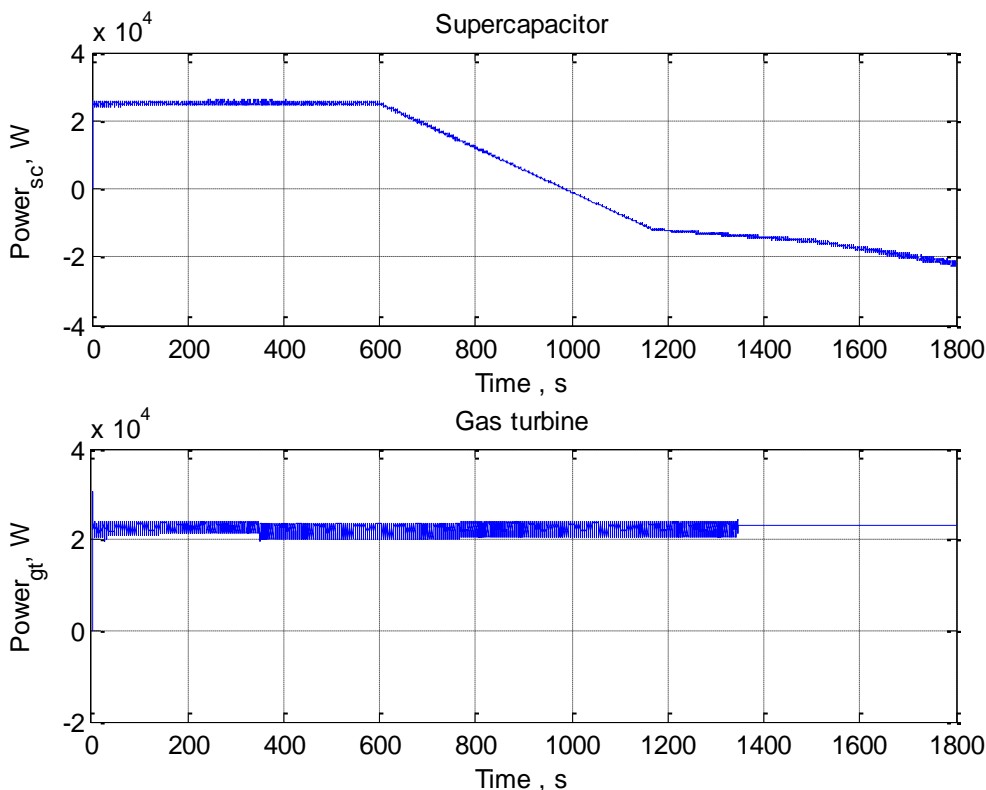

**Figure 9.** Realized power of supercapacitor and gas turbine at a minimum value of SC = 481F.

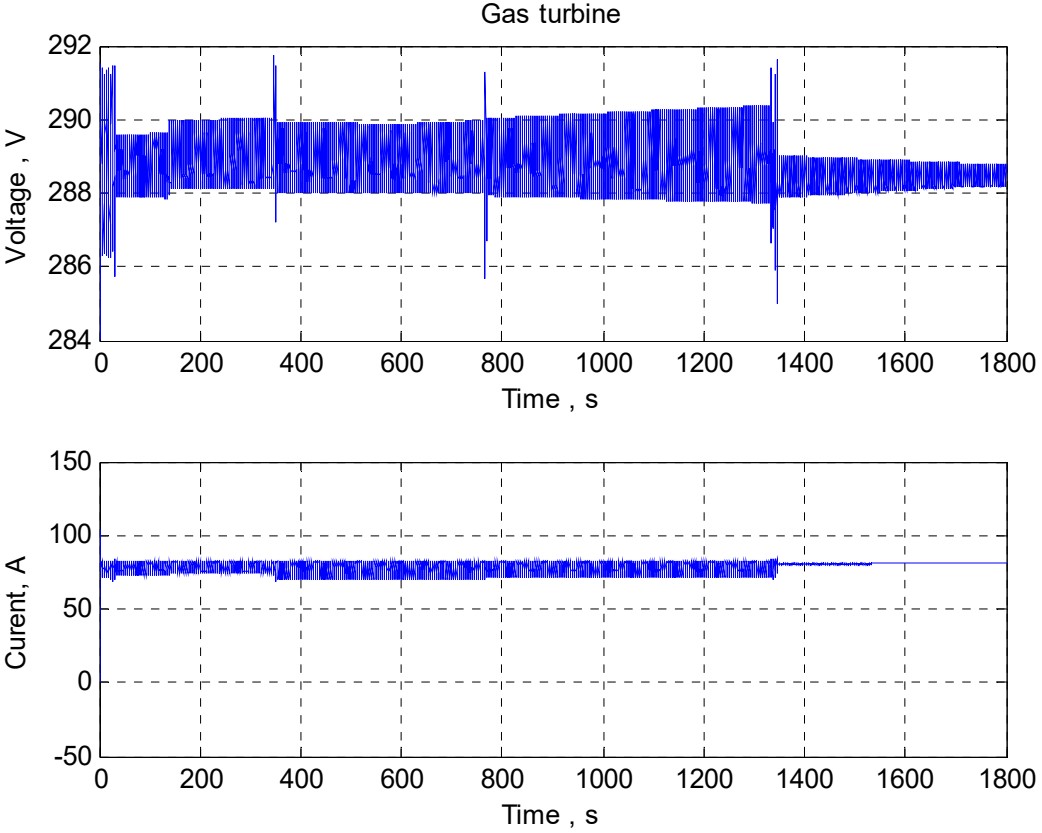

**Figure 10.** Voltages and current of gas turbine.

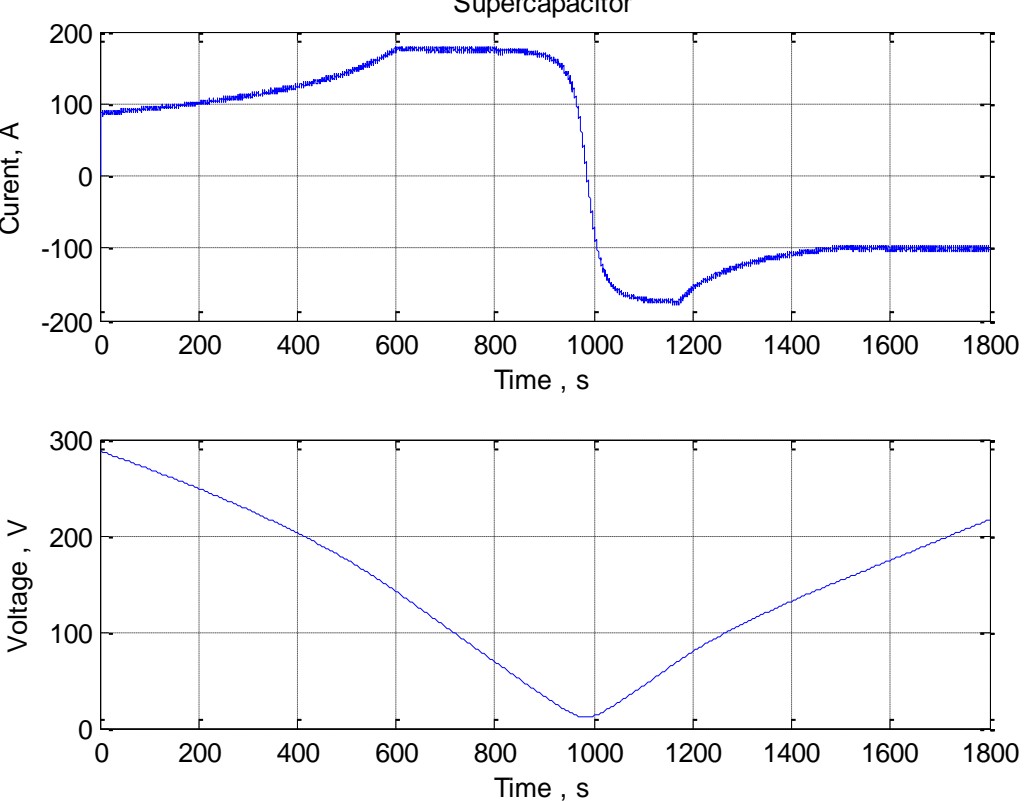

**Figure 11.** Voltages and current of supercapacitor.

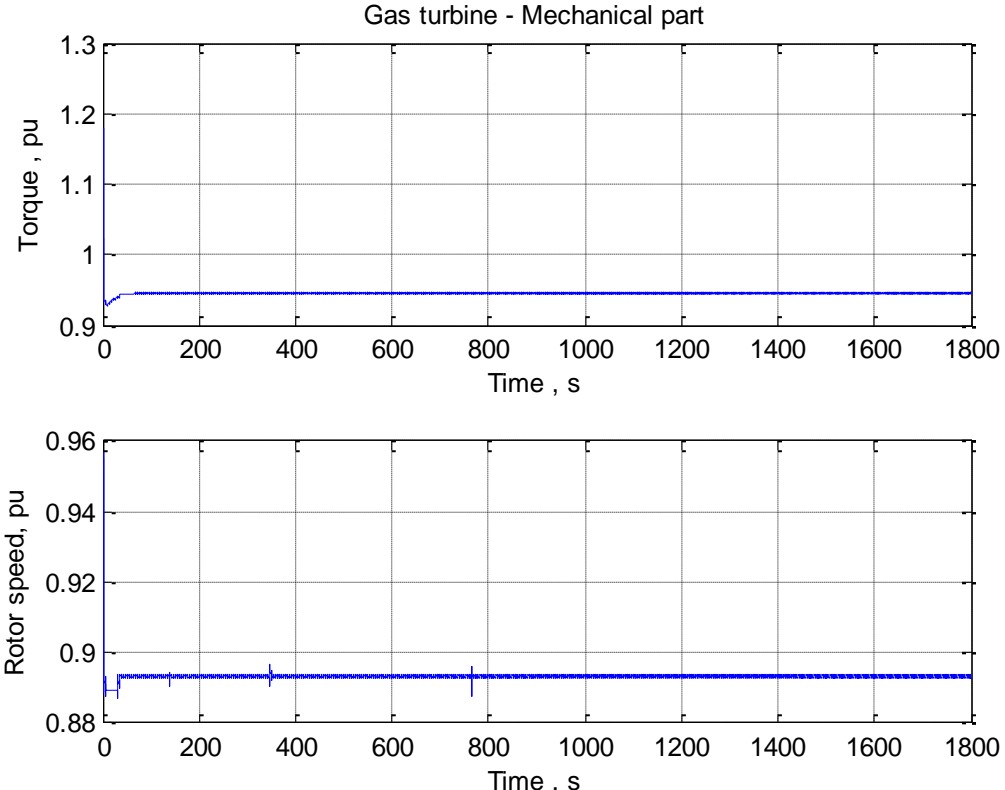

**Figure 12.** Mechanical characteristics of the turbine.

The results shown in Figure 9 give the distribution of the powers supplied to the load, establishing that the MGT generates the constant component of the power, and the SC the variable one. In this way, the desired load schedule to achieve a fast charge is realized. In Figure 10, from top to bottom, the output voltage of the MGT driven generator and the output current are given. It can be seen that even using only the built-in mechanical regulator MGT, stable values of the generated power are achieved.

In Figure 11, the current and voltage of the supercapacitor battery are also given. Initially, the supercapacitor adds its discharge current to the load until about 980 s and its voltage becomes almost zero (bottom graph). Then, according to the selected load graph in Figure 5, the excess energy generated by the MGT is used for charging of the supercapacitor and thus prepares the system for the next charge cycle.

Figure 12 shows the results of the simulation of the mechanical part of the MGT. From the analysis, it is found that the mechanical turbine governor provides a stable torque (upper graph) at a constant turbine speed (lower graph).

The presented graphical results confirm the workability of the system and make it possible to determine the values of all electrical and mechanical quantities through model-based design, and thus to make an optimal choice regarding topologies of power electronic devices and circuit elements. The comparison of these results with relevant studies given in [35–41] shows that there is a good match and in practice they can be used for the realization of a prototype, which is of interest to a Bulgarian company, a manufacturer of charging stations.

## 5. Conclusions

This manuscript presents a model-based design of an autonomous electric vehicle charging station based on MGT and a supercapacitor and its power flow control system. The use of a supercapacitor in the implemented system results in providing the required current through the load and helps to compensate for either the shortage or excess of MGT-generated energy during the charging process of an electric vehicle. In order to

find the minimum value of the supercapacitor, a series of simulations and an optimization procedure were carried out. Due to the low ESR of supercapacitors, they can deliver or store peak power during charge acceleration and deceleration processes, resulting in reduced system losses and increased overall efficiency. The control system that regulates the energy of the gas turbine and the supercapacitor is supposed in order to ensure that a certain charge cycle has been synthesized.

The use of an energy storage element helps to keep the microgas turbine operating close to rated power, where it is most efficient. The use of model-based design provides optimal results in terms of set ranges of input voltage, output voltage, operating frequency, load current ripple, and output voltage ripple, ultimately leading to the determination of the operating modes of power electronic converters with lower losses. The obtained simulation results clearly show that the developed system is a promising autonomous power supply option for the isolated operation of an electric vehicle charging station.

The use of the proposed approach for the implementation of charging stations is justified in most practical cases. For example, a large number of gas stations offer, in addition to gasoline and diesel fuel, natural gas. Natural gas can be used as a fuel for electricity generation through MGT while using existing infrastructure. In addition, the heat released during the generation is used for heating the gas station itself and the adjacent and additional facilities, e.g., shops, restaurants, recreation, etc., in the winter or cooling in the summer. In this way, the integration of the charging stations and the existing gas stations is carried out, especially given the transition between the different types of vehicles will be smooth and for a long enough time it will be necessary to offer all types of fuel. This would help saturate the highways with charging infrastructure, especially since it takes significant investment and time to build, which would inevitably affect the cost of charging with electricity. In addition, the released heat could also be used for the production of green vegetables in greenhouses and other energy-intensive industries.

**Author Contributions:** B.G., M.A., N.H. and G.A. were involved in the full process of producing this paper, including conceptualization, methodology, modeling, validation, visualization, and preparation of the manuscript. All authors have read and agreed to the published version of the manuscript.

**Funding:** This research was funded by Bulgarian National Scientific Fund, grant number КП-06-H57/7/16.11.2021, and the APC was funded by КП-06-H57/7/16.11.2021.

**Institutional Review Board Statement:** Not applicable.

**Informed Consent Statement:** Not applicable.

**Data Availability Statement:** Not applicable.

**Acknowledgments:** This research was carried out within the framework of the project "Artificial Intelligence-Based modeling, design, control and operation of power electronic devices and systems", КП-06-H57/7/16.11.2021, Bulgarian National Scientific Fund.

**Conflicts of Interest:** The authors declare no conflict of interest.

**Appendix A**

**Table A1.** Parameter values for micro gas turbine model.

| Parameter | Description | Value |
|:---:|:---:|:---:|
| W | Speed Controller | 25, p.u. |
| X | Lead Constant | 0.1, s |
| Y | Lag Constant | 0.05, s |
| Z | Mode = 1 (1 = droop, 0 = isochronous) | 1 |
| MAX | Fuel Demand Signal Upper Limit | 1.5, p.u. |
| MIN | Fuel Demand Signal Lower Limit | −0.1, p.u. |
| a | Valve Positioner | 1, s |

**Table A1.** *Cont.*

| Parameter | Description | Value |
|:---:|:---:|:---:|
| b | Valve Positioner | 0.05, s |
| c | Valve Positioner | 1, s |
| Wmin | No Load Fuel Consumption | 0.23 |
| TF | Fuel System Time Constant | 04, s |
| KF | Fuel System External Feedback Loop Gain | 0 |
| ECR | Delay of Combustion System | 0.01, s |
| ETD | Transport Delay of Turbine and Exhaust System | 0.02, s |
| TCD | Compressor Discharge Lag Time Constant | 0.2, s |
| TR | Rated Exhaust Temperature | 950, degree F |
| TT | Temperature Controller Integration Constant | 527, degree F |

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
