# Peer review of "Modeling and Simulation of a Low-Cost Fast Charging Station Based on a Micro Gas Turbine and a Supercapacitor"

_energies, doi:10.3390/en15218020_

Round 1
Reviewer 1 Report
The article presents the approach of modelling of the fast charging station based on MGT and supercapacitor. In my opinion the article has serious flaws and should be rejected. The most important are:
1. The article, methodology and results are presented in a chaotic non organized style. Some parts of the introduction present much information without references, while in some parts of the manuscript there are 15 references to just one sentence. It is very difficult to follow the ideas due to much unimportant information (e.g. how exhausting system works - it was not even taken into account - so why autors spend time providing information about mix of petrols etc.?). The introduction does not provide any state of the art regarding to the proposed models.
2. While autors spend much room for unimportant and excessive information, the presented models (especially equations related to them) are not provided or are not described anywhere. Especially Simulink models or results in fig. 7-12.
3. I do not see any novelty regarding to this article. All presented models of PMSM and system using PMSM is well known in the literature. I have also no idea of why authors provide fig. 6 while buck/boost model is a basics of electronics.
4. The authors did not analyse the viability of the proposed construction. Actually, to the best of my knowledge, there is no existing supercapacitor (or even cluster of capacitors) whose capacitance is equal to the proposed 480 F at as high voltage of almost 200 V (as presented in fig. 11). In my opinion there is a mistake in the authors methodology: That is why we use buck/boost regulator to reduce this value of the voltage into the resonable range. In this case, I have no idea why authors do not use this buck/boost unit (even previously described by them).
Some details:
Lines 43-57: It looks more like novel rather than scientific text. Please stay focused on the most important state of the art regarding to your work, providing references to the literature.
The introduction between lines 43 and 101 should be significantly shorten (by half or even more). There are much information without references and not related to the modelling. If I understand correctly, the goal of the article is modelling so the introduction about MGT should be max 40 lines and the rest of the Introduction should be state of the art regarding to its modelling.
Line 113: I do not understand the purpose of references [8-18].
Line 119: There should be reference to this turbine.
Line 146": Again, I do not understand the purpose of references [13-20]. You should explain them one by one not just giving them suddenly without any explanation.
Fig. 2: The quality of this picture is very low.
Line 159: There must be reference to Rowen's model.
Line 185: Again and again: "The output of the selector with the lowest values (called VCE) is the lowest from the three inputs and results in the least amount of fuel to the compressor-turbine [5-20]". - could you please explain, what is the purpose of giving 15 references in this place, without any information about what they present here?
Lines 230-233: unit cannot be italic.
From page 7, I do not understand what is happening in this manuscript (and research). Schamatics from fig. 3, 5 and 6 are described nowhere.
Figure 4 is illegible.
Figure 5 is illegible.
Figure 6. What is the reason of showing buck/boost converter? It is basics of electronics, there is nothing new.
Line 287 and the rest of the article, variables must be italic, otherwise it is not possible to distinquish them from the text.
Figures 7-12: the results presented here are described in nowhere... I have no idea what they present. Moreover, what is the purpose of showing empty parts of the figures (e.g. fig. 10, current below 0 A)?
Author Response
First of all, we would like to thank you for the thorough review of our paper (energies-1969460) and the useful remarks to improve it.
Reviewer 1
Comments to the Authors
The article presents the approach of modelling of the fast charging station based on MGT and supercapacitor. In my opinion the article has serious flaws and should be rejected. The most important are:
- The article, methodology and results are presented in a chaotic non organized style. Some parts of the introduction present much information without references, while in some parts of the manuscript there are 15 references to just one sentence. It is very difficult to follow the ideas due to much unimportant information (e.g. how exhausting system works - it was not even taken into account - so why autors spend time providing information about mix of petrols etc.?). The introduction does not provide any state of the art regarding to the proposed models.
- While autors spend much room for unimportant and excessive information, the presented models (especially equations related to them) are not provided or are not described anywhere. Especially Simulink models or results in fig. 7-12.
- I do not see any novelty regarding to this article. All presented models of PMSM and system using PMSM is well known in the literature. I have also no idea of why authors provide fig. 6 while buck/boost model is a basics of electronics.
- The authors did not analyse the viability of the proposed construction. Actually, to the best of my knowledge, there is no existing supercapacitor (or even cluster of capacitors) whose capacitance is equal to the proposed 480 F at as high voltage of almost 200 V (as presented in fig. 11). In my opinion there is a mistake in the authors methodology: That is why we use buck/boost regulator to reduce this value of the voltage into the resonable range. In this case, I have no idea why authors do not use this buck/boost unit (even previously described by them).
Some details:
- Lines 43-57: It looks more like novel rather than scientific text. Please stay focused on the most important state of the art regarding to your work, providing references to the literature.
- The introduction between lines 43 and 101 should be significantly shorten (by half or even more). There are much information without references and not related to the modelling. If I understand correctly, the goal of the article is modelling so the introduction about MGT should be max 40 lines and the rest of the Introduction should be state of the art regarding to its modelling.
- Line 113: I do not understand the purpose of references [8-18].
- Line 119: There should be reference to this turbine.
- Line 146": Again, I do not understand the purpose of references [13-20]. You should explain them one by one not just giving them suddenly without any explanation.
- 2: The quality of this picture is very low.
- Line 159: There must be reference to Rowen's model.
- Line 185: Again and again: "The output of the selector with the lowest values (called VCE) is the lowest from the three inputs and results in the least amount of fuel to the compressor-turbine [5-20]". - could you please explain, what is the purpose of giving 15 references in this place, without any information about what they present here?
- Lines 230-233: unit cannot be italic.
- From page 7, I do not understand what is happening in this manuscript (and research). Schamatics from fig. 3, 5 and 6 are described nowhere.
- Figure 4 is illegible.
- Figure 5 is illegible.
- Figure 6. What is the reason of showing buck/boost converter? It is basics of electronics, there is nothing new.
- Line 287 and the rest of the article, variables must be italic, otherwise it is not possible to distinquish them from the text.
- Figures 7-12: the results presented here are described in nowhere... I have no idea what they present. Moreover, what is the purpose of showing empty parts of the figures (e.g. fig. 10, current below 0 A)?.
To Reviewer 1:
Thank you very much for your review and valuable remarks.
- The article, methodology and results are presented in a chaotic non organized style. Some parts of the introduction present much information without references, while in some parts of the manuscript there are 15 references to just one sentence. It is very difficult to follow the ideas due to much unimportant information (e.g. how exhausting system works - it was not even taken into account - so why autors spend time providing information about mix of petrols etc.?). The introduction does not provide any state of the art regarding to the proposed models.
- I completely agree with your comment. The entire manuscript has been reviewed and substantially revised. We hope that the aim and main results are now presented in a better way and that the manuscript will be useful to readers.
- While autors spend much room for unimportant and excessive information, the presented models (especially equations related to them) are not provided or are not described anywhere. Especially Simulink models or results in fig. 7-12.
- I completely agree with your comment. The manuscript has been revised emphasizing the main goal of the manuscript - the presentation of a concept for the implementation of an autonomous charging station.
- I do not see any novelty regarding to this article. All presented models of PMSM and system using PMSM is well known in the literature. I have also no idea of why authors provide fig. 6 while buck/boost model is a basics of electronics.
- Thank you very much for your comment. The unique contribution of the manuscript is that based on existing models, a model of the overall system is achieved, together with the controller. In fig. 6, a bi-directional DC/DC converter is presented, and the innovative thing is the way of its control.
- The authors did not analyse the viability of the proposed construction. Actually, to the best of my knowledge, there is no existing supercapacitor (or even cluster of capacitors) whose capacitance is equal to the proposed 480 F at as high voltage of almost 200 V (as presented in fig. 11). In my opinion there is a mistake in the authors methodology: That is why we use buck/boost regulator to reduce this value of the voltage into the resonable range. In this case, I have no idea why authors do not use this buck/boost unit (even previously described by them).
- Thank you very much for your comment. An existing supercapacitor battery is specified in the rework. Also, a correction has been made to the system description in the part that deals with DC/DC converters, as the original version was unclear and confusing.
Some details:
- Lines 43-57: It looks more like novel rather than scientific text. Please stay focused on the most important state of the art regarding to your work, providing references to the literature.
- The manuscript has been completely revised, according to the recommendations and comments of all reviewers and the editor.
- The introduction between lines 43 and 101 should be significantly shorten (by half or even more). There are much information without references and not related to the modelling. If I understand correctly, the goal of the article is modelling so the introduction about MGT should be max 40 lines and the rest of the Introduction should be state of the art regarding to its modelling.
- Thank you very much for the recommendation. The manuscript has been completely revised and edited.
- Line 113: I do not understand the purpose of references [8-18].
- Fixed, thanks for the note.
- Line 119: There should be reference to this turbine.
- Fixed, thanks for the note.
- Line 146": Again, I do not understand the purpose of references [13-20]. You should explain them one by one not just giving them suddenly without any explanation.
- Fixed, thanks for the note.
- 2: The quality of this picture is very low.
- All figures have been re-examined, replacing low quality ones with new ones.
- Line 159: There must be reference to Rowen's model.
- Fixed, thanks for the note.
- Line 185: Again and again: "The output of the selector with the lowest values (called VCE) is the lowest from the three inputs and results in the least amount of fuel to the compressor-turbine [5-20]". - could you please explain, what is the purpose of giving 15 references in this place, without any information about what they present here?
- A very relevant remark! Corrected.
- Lines 230-233: unit cannot be italic.
- Fixed, thanks for the note.
- From page 7, I do not understand what is happening in this manuscript (and research). Schamatics from fig. 3, 5 and 6 are described nowhere.
- A very relevant remark! Corrected.
- Figure 4 is illegible.
- All figures have been re-examined, replacing low quality ones with new ones.
- Figure 5 is illegible.
- All figures have been re-examined, replacing low quality ones with new ones.
- Figure 6. What is the reason of showing buck/boost converter? It is basics of electronics, there is nothing new.
- Made a correction to the system description in the part that deals with DC/DC converters, as the original version was unclear and confusing. The new thing that is highlighted in the description is the energy flow management system.
- Line 287 and the rest of the article, variables must be italic, otherwise it is not possible to distinquish them from the text.
- Fixed, thanks for the note.
- Figures 7-12: the results presented here are described in nowhere... I have no idea what they present. Moreover, what is the purpose of showing empty parts of the figures (e.g. fig. 10, current below 0 A)?.
- Fixed, thanks for the note.
Thank you very much for your remarks and comments! They were very useful for us to emphasize the main tasks and contributions of the manuscript, and also to focus the attention of the readers on the new and original elements.
Reviewer 2 Report
This paper presents a model in Matlab/Simulink of a charging station supported by micro gas turbines. In my opinion, this work needs major amendments before considering it for publication in Energies. Please find my comments below:
· English must be notably enhanced and polished.
· In general, resolution of figures must be increased. Actually, some figures are ineligible (e.g. Fig. 2).
· Please unify the reference format to be coherent.
· Literature survey is scarce and ignores some recent articles in the field of charging stations modelling (e.g. 10.1016/j.energy.2022.124219).
· Parameters used in simulations (especially those reported in Appendix A), should be referenced.
· What is the importance of supercapacitor in the proposed charging mode? Why not use other storage technologies?
· System contributions seem poor and vague. The authors should list the actual novelties of this paper with respect others.
· I am unable to find any advantage in using micro gas turbines instead of other clean technologies like PV. The authors should discuss this point.
· More results should be provided comparing the developed model with other well-known approaches.
Thanks to the authors for your effort and time.
Author Response
Reviewer 2
Comments to the Authors
This paper presents a model in Matlab/Simulink of a charging station supported by micro gas turbines. In my opinion, this work needs major amendments before considering it for publication in Energies. Please find my comments below:
- English must be notably enhanced and polished.
- In general, resolution of figures must be increased. Actually, some figures are ineligible (e.g. Fig. 2).
- Please unify the reference format to be coherent.
- Literature survey is scarce and ignores some recent articles in the field of charging stations modelling (e.g. 10.1016/j.energy.2022.124219).
- Parameters used in simulations (especially those reported in Appendix A), should be referenced.
- What is the importance of supercapacitor in the proposed charging mode? Why not use other storage technologies?
- System contributions seem poor and vague. The authors should list the actual novelties of this paper with respect others.
- I am unable to find any advantage in using micro gas turbines instead of other clean technologies like PV. The authors should discuss this point.
- More results should be provided comparing the developed model with other well-known approaches.
Thanks to the authors for your effort and time.
To Reviewer 2:
Thank you for your review and valuable remarks.
- English must be notably enhanced and polished.
- Thank you very much for the recommendation. The manuscript has been completely revised and edited.
- In general, resolution of figures must be increased. Actually, some figures are ineligible (e.g. Fig. 2).
- All figures have been re-examined, replacing low quality ones with new ones.
- Please unify the reference format to be coherent.
- Fixed, thanks for the note.
- Literature survey is scarce and ignores some recent articles in the field of charging stations modelling (e.g. 10.1016/j.energy.2022.124219).
- Thank you very much for the recommendation. The manuscript has been completely revised and edited, with the literature updated and the literature review rewritten.
- Parameters used in simulations (especially those reported in Appendix A), should be referenced.
- Fixed, thanks for the note.
- What is the importance of supercapacitor in the proposed charging mode? Why not use other storage technologies?
- The role of the supercapacitor is to provide the required load schedule. Chosen with this particular type of energy storage cell due to its ability to deep discharge, numerous charge and discharge cycles, and its better efficiency compared to other types of batteries. The disadvantage is its relatively high price, and the other elements of the system are not as cheap as possible.
- System contributions seem poor and vague. The authors should list the actual novelties of this paper with respect others.
- In the conclusion section, the contributions are motivated and indicated.
- I am unable to find any advantage in using micro gas turbines instead of other clean technologies like PV. The authors should discuss this point.
- This aspect was also discussed during the editing of the manuscript. Thank you very much for the remark!
- More results should be provided comparing the developed model with other well-known approaches.
- During the revision of the manuscript, this point was also commented on.
Thank you very much for your remarks and comments! They were very useful for us to emphasize the main tasks and contributions of the manuscript, and also to focus the attention of the readers on the new and original elements.
Reviewer 3 Report
The article contains an interesting idea about the use of a gas microturbine and a supercapacitor in an electric vehicle charging station. However, the style of presentation of information and the design of the article have major drawbacks.
The Introduction section redundantly provides information on the advantages of gas microturbines. At the same time, data on the use of hybrid systems for charging electric vehicles are not provided.
The description of Figure 1 does not correspond to the diagram presented on it.
Links to Figures 2, 3 and other Figures are broken.
The inscriptions in Figure 2 are hard to read.
In line 172, the meaning of the phrase “upper and lower fuel limits” is not clear.
It is not clear why mode 2 is being investigated in detail.
Line 274 says “two thyristors”. There are no thyristors in the diagram.
Section 4 shows the simulation results for mode 2 for a power of 22 kW. However, Figure 4 shows a power of 6.6 kW.
The simulation results are poorly commented.
Author Response
Reviewer 3
Comments to the Authors
- The article contains an interesting idea about the use of a gas microturbine and a supercapacitor in an electric vehicle charging station. However, the style of presentation of information and the design of the article have major drawbacks.
- The Introduction section redundantly provides information on the advantages of gas microturbines. At the same time, data on the use of hybrid systems for charging electric vehicles are not provided.
- The description of Figure 1 does not correspond to the diagram presented on it.
- Links to Figures 2, 3 and other Figures are broken.
- The inscriptions in Figure 2 are hard to read.
- In line 172, the meaning of the phrase “upper and lower fuel limits” is not clear.
- It is not clear why mode 2 is being investigated in detail.
- Line 274 says “two thyristors”. There are no thyristors in the diagram.
- Section 4 shows the simulation results for mode 2 for a power of 22 kW. However, Figure 4 shows a power of 6.6 kW.
- The simulation results are poorly commente.
To Reviewer 3:
Thank you for your review and valuable remarks.
- The article contains an interesting idea about the use of a gas microturbine and a supercapacitor in an electric vehicle charging station. However, the style of presentation of information and the design of the article have major drawbacks.
- Thank you very much for your comment! The manuscript has been completely revised and edited.
- The Introduction section redundantly provides information on the advantages of gas microturbines. At the same time, data on the use of hybrid systems for charging electric vehicles are not provided.
- Thank you very much for the recommendation. The manuscript has been completely revised and edited, with the literature updated and the literature review rewritten.
- The description of Figure 1 does not correspond to the diagram presented on it.
- Fixed, thanks for the note.
- Links to Figures 2, 3 and other Figures are broken.
- Fixed, thanks for the note.
- The inscriptions in Figure 2 are hard to read.
- Fixed, thanks for the note.
- In line 172, the meaning of the phrase “upper and lower fuel limits” is not clear.
- The maximum and minimum value of the fuel supplied to the turbine is taken into account. The text has been edited.
- It is not clear why mode 2 is being investigated in detail.
- In our opinion, it is the most common in practice. The model-based design approach used makes it possible to explore all possible operating modes.
- Line 274 says “two thyristors”. There are no thyristors in the diagram.
- Fixed, thanks for the note.
- Section 4 shows the simulation results for mode 2 for a power of 22 kW. However, Figure 4 shows a power of 6.6 kW.
- Fixed, thanks for the note.
- The simulation results are poorly commente.
- Thank you very much for your comment! When editing the manuscript, a description of the results was added.
Thank you very much for your remarks and comments! They were very useful for us to emphasize the main tasks and contributions of the manuscript, and also to focus the attention of the readers on the new and original elements.
Round 2
Reviewer 1 Report
The article has been rebuilt and now it is more clear. I have still a few questions/suggestions:
I still do not understand the goal of this Introduction. MGT is just a part of the whole system, so why do you spend almost whole Introduction onto MGT?
"The main purpose of the manuscript is to study a system for fast charging of electric vehicles which is autonomous from the power supply network..." - from the manuscript the reader still does not know anything about the state of the art regarding to modelling of similar systems nor other autonomus systems currently being in use/under consideration. In general, the state of the art is missing.
Why some numbers are in italic?
Figures: some names start from capital letters some from small. Please unify it.
Chapter 2.2: You provide many equations however their source is missing. I am not sure if the Table 1 is needed. I still do not understand, why some units are written in italic style. You provide many values such as resistance or inductance, however I have no idea from where do you have them.
Line 281: PMSG?
The proposed system requires validation by experiment or by other theoretical method.
Author Response
First of all, we would like to thank you for the thorough review of our paper (energies-1969460) and the useful remarks to improve it.
Reviewer 1
Comments to the Authors
The article has been rebuilt and now it is more clear. I have still a few questions/suggestions:
I still do not understand the goal of this Introduction. MGT is just a part of the whole system, so why do you spend almost whole Introduction onto MGT?
"The main purpose of the manuscript is to study a system for fast charging of electric vehicles which is autonomous from the power supply network..." - from the manuscript the reader still does not know anything about the state of the art regarding to modelling of similar systems nor other autonomus systems currently being in use/under consideration. In general, the state of the art is missing.
Why some numbers are in italic?
Figures: some names start from capital letters some from small. Please unify it.
Chapter 2.2: You provide many equations however their source is missing. I am not sure if the Table 1 is needed. I still do not understand, why some units are written in italic style. You provide many values such as resistance or inductance, however I have no idea from where do you have them.
Line 281: PMSG?
The proposed system requires validation by experiment or by other theoretical method.
To Reviewer 1:
Thank you very much for your review and valuable remarks.
I still do not understand the goal of this Introduction. MGT is just a part of the whole system, so why do you spend almost whole Introduction onto MGT?
- Thank you very much for your comment. The introduction has been expanded and the section on MGT has been reduced. Our idea was to present all the advantages of this source of energy and therefore we have given it current attention.
- "The main purpose of the manuscript is to study a system for fast charging of electric vehicles which is autonomous from the power supply network..." - from the manuscript the reader still does not know anything about the state of the art regarding to modelling of similar systems nor other autonomus systems currently being in use/under consideration. In general, the state of the art is missing.
- Thank you very much for the remark. A review of other loading systems has also been added, thus improving the manuscript and more convincingly demonstrating the need for such research.
- Why some numbers are in italic?
- Text formatting error. Corrected.
- Figures: some names start from capital letters some from small. Please unify it.
- Text formatting error. Corrected.
- Chapter 2.2: You provide many equations however their source is missing. I am not sure if the Table 1 is needed. I still do not understand, why some units are written in italic style. You provide many values such as resistance or inductance, however I have no idea from where do you have them.
- Thank you very much for the question. The source by which these parameters are defined has been added during the rework.
- The introduction between lines 43 and 101 should be significantly shorten (by half or Line 281: PMSG?
- Thank you very much for the recommendation. The symbols used are unified.
Thank you very much for your remarks and comments! They were very useful for us to emphasize the main tasks and contributions of the manuscript, and also to focus the attention of the readers on the new and original elements.
Reviewer 2 Report
Dear authors,
the paper has been notably improved and therefore it is now ready for publication.
Author Response
Thank you very much for your remarks and comments! They were very useful for us to emphasize the main tasks and contributions of the manuscript, and also to focus the attention of the readers on the new and original elements.
Reviewer 3 Report
The authors of the manuscript carefully completed the work on the comments. As a result, the style of presentation, the quality of illustrations have become much better. The study results are well commented.
Author Response

(The authors gave the same response as above.)

Round 3
Reviewer 1 Report
The manuscript has been again revised. In spite of low novelty, it contains a little useful information. It can be accepted in the present form.